# Auditory and Visual Response Inhibition in Children with Bilateral Hearing Aids and Children with ADHD

**DOI:** 10.3390/brainsci10050307

**Published:** 2020-05-18

**Authors:** Laura Bell, Wolfgang Scharke, Vanessa Reindl, Janina Fels, Christiane Neuschaefer-Rube, Kerstin Konrad

**Affiliations:** 1Child Neuropsychology Section, Department of Child and Adolescent Psychiatry, Psychosomatics and Psychotherapy, Medical Faculty, RWTH Aachen University, D-52074 Aachen, Germany; Wolfgang.Scharke@psych.rwth-aachen.de (W.S.); vreindl@ukaachen.de (V.R.); kkonrad@ukaachen.de (K.K.); 2Institute of Cognitive and Experimental Psychology, RWTH Aachen University, D-52074 Aachen, Germany; 3JARA-Brain Institute II, Molecular Neuroscience and Neuroimaging (INM-11), RWTH Aachen & Research Centre Juelich, D-52428 Juelich, Germany; 4Teaching and Research Area of Medical Acoustics, Institute of Technical Acoustics, RWTH Aachen University, D-52074 Aachen, Germany; jfe@akustik.rwth-aachen.de; 5Clinic of Phoniatrics, Pedaudiology, and Communication Disorders, Medical Faculty, RWTH Aachen University, D-52074 Aachen, Germany; cneuschaefer@ukaachen.de

**Keywords:** response inhibition, hearing loss, hearing aids, ADHD, modality-specific, supra-modal, go/nogo

## Abstract

Children fitted with hearing aids (HAs) and children with attention deficit/hyperactivity disorder (ADHD) often have marked difficulties concentrating in noisy environments. However, little is known about the underlying neural mechanism of auditory and visual attention deficits in a direct comparison of both groups. The current functional near-infrared spectroscopy (fNIRS) study was the first to investigate the behavioral performance and neural activation during an auditory and a visual go/nogo paradigm in children fitted with bilateral HAs, children with ADHD and typically developing children (TDC). All children reacted faster, but less accurately, to visual than auditory stimuli, indicating a sensory-specific response inhibition efficiency. Independent of modality, children with ADHD and children with HAs reacted faster and tended to show more false alarms than TDC. On a neural level, however, children with ADHD showed supra-modal neural alterations, particularly in frontal regions. On the contrary, children with HAs exhibited modality-dependent alterations in the right temporopolar cortex. Higher activation was observed in the auditory than in the visual condition. Thus, while children with ADHD and children with HAs showed similar behavioral alterations, different neural mechanisms might underlie these behavioral changes. Future studies are warranted to confirm the current findings with larger samples. To this end, fNIRS provided a promising tool to differentiate the neural mechanisms underlying response inhibition deficits between groups and modalities.

## 1. Introduction

### 1.1. Modality-(In)Dependent Response Inhibition

Cognitive control develops continuously throughout childhood, and response inhibition plays an important role in ensuring that only appropriate responses are given [1,2,3]. The maturation of cognitive abilities, however, strongly relies on the perception, exploration and integration of sensory information that is provided in our complex everyday environment [4,5,6]. While information that is presented redundantly across senses can be promoted, and conflicting information impairs response inhibition efficiency in typically developing (TD) individuals [7], each sensory modality carries a domain specific dominance and thus offers an individual informative value. Recently, the importance of perceptual processes has also been reported for response inhibition mechanisms in TD adults between the visual and tactile modality [8]. The tactile modality was shown to be more efficient than the visual modality in triggering successful response inhibition [8]. Furthermore, auditory attention has been shown to involve the representation of sound in the auditory cortex, especially secondary regions, as well as cortical areas associated with visual attention control [9]. The engagement of similar neural networks in auditory and visual paradigms might suggest a supra-modal organization of attention. However, the utilization of these networks, i.e., the exact brain regions within the networks and the extent of their contribution, appears to vary according to the stimulus modality [9]. Behavioral and neural response inhibition mechanisms might therefore depend on the stimulus modality in a similar way.

### 1.2. Modality-(In)Dependent Response Inhibition in Auditory Deprivation

In addition, given that our everyday environment typically consists of complex multisensory features, and considering that audiovisual integration scaffolds cognitive development beyond language perception [4,10,11], it is reasonable to assume that the deprivation of one sense has far-reaching consequences. Indeed, it has been shown that, e.g., early auditory deprivation, leads to various behavioral and neural alterations [12,13,14]. For example, the orienting response of individuals with hearing loss (HL) was shown to be attenuated, as evidenced by abrupt changes in orientation movements [15]. Notably, attention capture is a multimodal phenomenon. While the human visual system is incapable of capturing the entire surroundings, auditory events can guide visual attention [16,17]. Human neuroimaging studies and animal models suggested that alterations after sensory deprivation might be the result of the lack of sensory experience and thus the missing scaffolding mechanism of the deprived sense for the remaining senses [18], as well as changes in network connectivity [19]. Hence, early auditory deprivation might affect processes beyond unimodal perception and the auditory cortices. Consistent with this idea, previous findings reported alterations following auditory deprivation in several cognitive abilities of the child. Alterations included impairments in speech and language, in executive functions and in behavioral regulation. The impairments, in turn, had a negative impact on school performance and the development of behavioral problems [20,21,22,23,24,25]. Comparably, behavioral alterations have been reported in response inhibition in HL [25]. In response inhibition paradigms such as the go/nogo task, the participant is required to respond quickly to a go stimulus/-i, while inhibiting responses to a nogo stimulus. Task performance is often assessed by means of reaction times (RTs), RT variability, false alarm (FA) rates, i.e., reacting to a nogo stimulus, and omission rates, i.e., not reacting to a go stimulus. A large longitudinal study reported higher FA rates in a visual go/nogo paradigm in children with severe to profound HL and aided hearing by means of cochlear implantation compared to typically developing children (TDC; [25]). The go/nogo task performance in turn predicted parent-reported behavioral problems [25]. However, so far, the behavioral and neural effects of early auditory deprivation on behavior regulation in children with HL are not fully understood. In particular, little is known about individuals with residual hearing capacity and children fitted with hearing aids (HAs). It has been suggested that children with HL show increased rates of behavioral disorders associated with attention deficits and impulsiveness, such as attention deficit/hyperactivity disorder (ADHD) or oppositional defiant disorder (ODD)/conduct disorder (CD; e.g., see, for a review and a meta-analysis, Theunissen, Rieffe, Netten, Briaire, Soede, Schoones and Frijns [26] and Stevenson, et al. [27]). Notably, although some studies have reported an increased risk of ADHD in individuals with HL, there is a large inconsistency in the reported prevalence rates [27,28]. A meta-analysis revealed that scores on behavioral difficulties assessments in children with HL are about 0.25 to 0.33 times the standard deviation higher than those of their TD peers [27]. However, it was suggested that the attention and response inhibition difficulties observed in children with HL are not necessarily linked to ADHD [27,29]. Thus, while behavior regulation difficulties are more commonly reported in children with HL in dimensional assessments of emotional and behavioral problems (e.g., through the Strengths and Difficulties Questionnaire (SDQ)), attention and impulsiveness that occur independent of an ADHD diagnosis may be of less severity than previously assumed, when considered dimensionally [27].

### 1.3. Modality-(In)Dependent Response Inhibition in ADHD

In ADHD, impairments of response inhibition are widely recognized as one core deficit of the diagnosis [30,31,32]. Alterations in activation patterns within prefrontal, parietal and striatal areas are commonly linked to response inhibition impairments [33,34,35,36]. Further, children with ADHD showed an attenuated deactivation of the default mode network (DMN) during response inhibition tasks [37]. Interestingly, in the context of the DMN, a supra-modal extrinsic task mode network has been proposed, that includes frontal, temporal and parietal regions [38]. Thus, supra-modal alterations within this task mode network, next to the DMN attenuations, appear plausible during response inhibition. While the response inhibition deficit in ADHD is well investigated [30,31,32], the majority of studies focus on visual response inhibition (for meta-analyses see Lijffijt, et al. [39] and Simmonds, et al. [40]). In visual response inhibition paradigms, children with ADHD showed a pattern of higher RT variability and elevated FA rates compared to TDC [41,42]. Although auditory noise has been shown to be beneficial under certain circumstances in ADHD [43], generally, auditory noise is well known to disturb cognitive performance across different populations [43,44,45], and it remains unclear how children with ADHD inhibit their behavior in response to auditory stimuli. Few studies that investigated auditory response inhibition found that children with ADHD (symptomatology) had response inhibition difficulties in both auditory and visual tasks [46,47]. In a previous study, for example, children with ADHD differed from TDC in RTs and omission rates in the auditory and visual modality [47]. In the visual task, however, children with ADHD showed additionally elevated FA rates. Thus, although attentional deficits across modalities appeared to be a key feature of ADHD, inattentive and impulsive behavior was more pronounced in the visual than the auditory task. This modality-specific effect has been hypothesized to underlie both the developmental delay in ADHD and the asynchronous maturation of hearing and vision [47].

### 1.4. Current Objectives

There is a need for studies utilizing simultaneous visual and auditory response inhibition paradigms to verify modality-(in)dependent mechanisms in TDC, and to clarify whether populations that exhibit or have an increased risk to develop behavior regulation difficulties show supra-modal or modality-specific deficits. To our knowledge, no study has directly compared auditory and visual response inhibition simultaneously in children with HL and children with ADHD. Disregarding the potentially different pathophysiology of response inhibition problems of both groups, this could provide insights into the variability of response inhibition difficulties across populations and sensory modalities. Therefore, the aim of the current functional near-infrared spectroscopy (fNIRS) study was the comparison of the behavioral performance and neural activity during an auditory and visual go/nogo paradigm of children with HL (not diagnosed with ADHD) and children with ADHD (and normal hearing) to TDC. FNIRS is a versatile and non-invasive imaging technique that is less susceptible to motion artifacts than functional magnetic resonance imaging (fMRI). It is especially adequate for investigations of children with HAs and auditory tasks, as it is a quiet optical imaging technique for which the contraindications of fMRI do not apply [48]. We expected behavioral difficulties in response inhibition in children with HL and children with ADHD that might depend on stimulus modality. Based on previous reports (e.g., [28]), less pronounced behavioral alterations were expected for children with HL, with difficulties being particularly present in the auditory condition. On the contrary, based on previous findings [47], response inhibition difficulties in ADHD were expected to be more prominent during the visual condition. Correspondingly, a differential functional disorganization of attention networks was anticipated in children with HL and children with ADHD, compared to TDC. Given the limited research on modality-specific and supra-modal neural differences of response inhibition in each group of interest, and considering that, to the best of our knowledge, no neuroimaging study has previously compared children with ADHD and children with HL, no a priori hypotheses on these neural differences between groups were made. Therefore, an exploratory, channel (Ch)-based, rather than a region-of-interest-based, approach was performed, to investigate potential modality-(in)dependent differences in neural activation patters across bilateral frontal, parietal and temporal brain regions between groups.

## 2. Materials and Methods

### 2.1. Participants

Fifteen children with a bilateral HL, wearing bilateral HAs (HA group), 20 children with a previously assigned clinical diagnosis of ADHD (ADHD group) and 27 TDC (TD group) between the age of 6–13 years (*M* = 9.84, *SD* = 1.90) were included in the current study. Parental informed consent and children’s assent was obtained, and children were reimbursed for study participation. The study was approved by the local ethical committee (EK 188/15) and conducted in accordance with the Code of Ethics of the World Medical Association (Declaration of Helsinki). An unaided pure-tone audiometry was performed for all children with HAs within three weeks of study participation (see Appendix A and Appendix A). Notably, children with HL performed the current paradigm wearing their own HAs. Normal hearing of the TD and ADHD group was based on the mandatory early hearing screening (U9, conducted by a pediatrician between 60 and 64 month of age, including headphone-based audiometry, and documented in the child’s medical record) as well as parental report on the day of testing. Participants were excluded if one of the following criteria applied: (i) a child in the HA or TD group was diagnosed with ADHD, or (ii) a child in any group was diagnosed with any other diagnosis than ADHD that may alter the neural organization, e.g., autism spectrum disorder. TDC were recruited through local schools and previous study participation. Children fitted with bilateral HAs were recruited through teacher-parent communications from a school offering special education with a main focus on hearing and communication. The recruitment of clinically referred children with a previously assigned clinical diagnosis of ADHD took place through the Outpatient Unit of the Department of Child and Adolescent Psychiatry, Psychosomatics and Psychotherapy at the University Hospital Aachen, Germany. Attention deficits and ADHD symptoms were assessed through the German ADHD rating scale for parents or teachers (Fremdbeurteilungsbogen für Hyperkinetische Störungen, FBB-HKS [49]; total: *n* = 56; missing: *n* = 6 (in the TD group); parental *n* = 52; teacher: *n* = 4 (in the HL group)) and the parental Child Behavior Checklist (CBCL [50]; total: *n* = 60, missing: *n* = 2 (1 TDC and 1 child with HL)). Children with ADHD treated with medication (e.g., methylphenidate) were asked to stop their medication 24 h before participation. All children had normal or corrected to normal vision. Based on the unequal gender distribution (*χ*^2^(1, *n* = 62) = 5.20, *p* = 0.07), we systematically controlled for an effect of gender in all behavioral analyses (Section 3.1). For the subsequent neural analyses (Section 3.2), a matching procedure was applied, due to the unequal gender distribution and samples sizes after fNIRS signal quality control (Section 2.4). The demographic characteristics are listed separately for the behavioral and neural analyses in Table 1.

### 2.2. Set-Up and Go/Nogo Paradigm

To ensure silence, the assessment took place within a sound insulated booth (L × W × H = 2.12 × 2.12 × 2 m; 9 m^3^; fulfilling the International Organization for Standardization requirements [51,52,53]). Within the booth, all children were seated in front and with a distance of 0.9 m to a screen. Two loudspeakers (Neumann KH-120A; Georg Neumann GmbH, Berlin, Germany) were positioned in the front edges of the booth and at average height of the children’s ear axis. Well-being and performance were monitored by a webcam and through a window. All children were assessed by a go/nogo paradigm with alternated auditory and visual conditions with a total of six blocks, i.e., three blocks for each condition (Figure 1; programmed in Presentation (Neurobehavioral Systems, Inc., Version 17.1. Berkeley, CA, USA), that were preceded by a short notice to indicate the sensory modality of the block. In the auditory condition, sinusoidal tones of different tone frequencies were delivered in pseudorandomized order at 70 dBA sound pressure level. The highest frequency (910 Hz) represented the nogo stimulus. All tones were normalized according to standards of the EBU-R128 [54]. In the visual condition, a pseudorandomized order of letters was displayed on the screen. The nogo stimulus was the letter “X”. All stimuli were presented for 350 ms and intertwined by a randomized inter-stimulus interval of, on average, 1525.5 ms. Children were required to respond via a button press with their right (dominant) hand, as fast and as accurately as possible, to the go stimuli. Initially, the children were introduced to the auditory stimuli to ensure audibility and correct differentiation of the tones. Thereupon, 2 × 2 exercise runs (an auditory and a visual block, consisting of 10-trials each) were completed to ensure the understanding of the procedure. While visual feedback was given during the first exercise run of each condition, the second exercise run simulated the testing condition (without any feedback). While the children performed the task, fNIRS recordings were obtained concurrently.

### 2.3. FNIRS

#### 2.3.1. Probe Set Configuration

The multi-Ch ETG4000 optical topography system (Hitachi Medical Corporation, Tokyo, Japan) was utilized to measure changes in oxygenated (∆HbO) and deoxygenated (∆HbR) hemoglobin concentrations. Measurements were recorded with a sampling frequency of 10 Hz and two wavelengths (695 and 830 nm). During the experiment, the children wore two × 3 × 5 probe sets that were placed symmetrically on each side of the head within an electroencephalography (EEG) cap (Easycap GmbH, Herrsching, Germany). A total of 44 measurement Chs with 22 per probe set was placed at a Ch-to-Ch-distance of 3 cm. The receiving optode of the lowest row of each probe set (left: optode 17, right: optode 26; as later illustrated in Figures 3 and 4) was positioned above the ear (above T3/T4 within the 10/20 system [55]), and the anterior corner was oriented towards the eyebrows. A 3D-digitizer-free virtual registration was applied for Ch-localization [56]. The Ch allocations are based on simulations, providing estimates of stereotactic brain coordinates [56]. The 2 × 3 × 5 fNIRS configuration resembled placements by the Jichi Medical University [57]. Herein, we refer to the anatomical brain labels of the highest probability.

#### 2.3.2. FNIRS Quality Control and Preprocessing

Metrics for signal quality control were the coefficients of variation (exclusion of HbO/HbR variations above 10% and exclusion of a variation difference between HbO and HbR exceeding 5% [58]) and exclusion of Chs with measurement error identified by the repetitive occurrence of the same value for 1 s, a flat line. The quality check was performed in MATLAB (R2019a, The MathWorks Inc., Natick, MA, USA), using self-written scripts. Due to poor overall quality (>40% of poor quality Chs), three children from the HA group, two children from the ADHD group and six children from the TD group were excluded from the neural data analyses. On average, 9.25% (4 out of 44 Chs) were removed across the remaining sample.

After quality control, the fNIRS data was preprocessed in MATLAB (R2019a, The MathWorks Inc., Natick, MA, USA) with self-written scripts and scripts of the HomER2 (Huppert, et al. [59]; version: homer2_src_v2_8_11022018) and SPM-fNIRS toolbox [60]. All steps of the preprocessing pipeline are depicted in Figure 2. A mean concentration with a time-window of 5 s after block onset, adjusting for the buildup of the hemodynamic response, to the end of the block (26 s), was utilized for the reported statistical analyses.

### 2.4. Matching Procedure

The exclusion of participants due to poor Ch-quality led to a wide variation in group size and gender distribution between groups for the neural analyses. To account for the variation in group size in the neural analyses, the sample sizes of the ADHD and TD group were reduced while avoiding an unequal gender and age distribution. That is, three further male participants from the ADHD group and one male participant from the TD group were excluded (for the final sample see Table 1). Due to its smallest sample size, gender matching was based on the gender distribution of the HA group. Perfect gender matching was not possible, due to the higher proportion of male children with ADHD. However, the gender difference between groups did not reach significance (*χ*^2^(2, *n* = 62) = 5.20, *p* = 0.07) and a higher prevalence of male compared to female children with ADHD is frequently observed in clinically referred samples [61].

### 2.5. Analyses

The analyses for behavioral and neural data were performed in R [62]. In order to check possible influential factors, linear mixed models (LMMs) were applied for the dependent variables (DVs) of behavioral performance in the go/nogo task and the neural activation using the R package *lme4* [63]. To obtain *p*-values of the fixed effects, the *anova* function of the *lmerTest* package was used [64]. The *R*^2^ beta was calculated using the R package *r2glmm* and the Kenward Roger approach [65]. Models were fitted using REML and the full model was always reported. For the behavioral analyses, with the sample of *n* = 62 subjects in total, the full model included group as 3-level (HA, ADHD, TD), and gender as 2-level between-subject factor, as well as condition (auditory, visual) as 2-level repeated within-subject factor. The interaction term of group-by-condition was included to explicitly evaluate whether the effect of condition varied between groups, or the effect of group between modalities. It should be noted that gender always remained in each model of the behavioral performance measures to take into account the unequal gender distribution between the groups. For the neural analyses, data analyses were conducted within the smaller sample consisting of *n* = 42 subjects, based on the gender matching procedure described in Section 2.4. The full model consisted only of the group as 3-level between-subject factor (HA, ADHD, TD) and the condition (auditory, visual) as 2-level repeated within-subject factor. Again, the interaction term of group-by-condition was included. Group-by-condition interactions were broken down by examining group effects per condition (2 LMMs) and condition effects per group (3 LMMs). Post-hoc comparisons of significant effects with the *emmeans* R package [66] were FDR corrected. For all analyses in this study, the level of significance was set at α = 0.05. *Partial eta-squared (η^2^)* was derived for the post-hoc comparisons of condition per group via the R package *lmSupport* [67] and for post hoc comparisons of group effects conducted with the *emmeans* function, *Cohen’s d* was derived.

For the behavioral data, separate independent LMMs were performed for the following DVs: FA rates (reactions to nogo stimuli; in percentage), mean RTs (RT_go_) of all correct Go trials (Hits) and response variability (standard deviation of RT_go_; SD_RTgo_)_._ Prior to the analyses, RTs were log-transformed and RTs below 100 ms were excluded. Additionally, the speed-accuracy trade-off (SAT = correlation of *z*-transformed FA-rate and MeanRT_go_), was analyzed by one-tailed Spearman’s rho, due to the expected negative relationship between RT and FA rates. Moreover, a study by van Belle, et al. [68] indicated that parameter estimates of the ex-Gaussian distribution [69], accounting for the positively skewed RT data, are superior in describing go/nogo task performance, as well as the behavioral characteristics of ADHD. RTs were therefore additionally investigated by the parameter estimates *mu* and *sigma* (i.e., mean and standard deviation of the normal distribution) as well as *tau* (i.e., mean and variance of the exponential distribution). The parameter estimates were determined by maximum likelihood estimation, using the R package *retimes* [70]. An iteration rate of 10,000 for bootstrapping was applied.

For the neural data, an LMM was run for each of the 44 Chs per chromophore (HbO, HbR). Due to the exploratory purpose of the current study, the smaller number of subjects included in the neural data analyses, and based on the decision to conduct Ch-wise comparisons of effects of condition, group and their interaction on activation patterns, we decided against correction for multiple comparisons (44 Chs). All neural analyses were therefore performed using LMMs with an uncorrected threshold of *p* < 0.05. Notably, post-hoc comparisons of group, condition and group-by-condition interactions were conducted identically to the behavioral analyses. Therefore, the post-hoc comparisons were FDR corrected.

FNIRS activation maps per group and condition were created in MATLAB (MATLAB Release 2019a. The MathWorks, Inc., Natick, MA, USA), using the modified *plotTopoMap* function by Cui [71]. The activation maps depict the contrasts of ΔHbO and ΔHbR in each Ch against baseline (uncorrected one-sample *t*-tests).

Finally, exploratory analyses to assess brain-behavior correlations were conducted. Spearman correlations were computed between the FA rate and neural activation per group (HA, ADHD, TD) and condition (auditory, visual). Chs showing significant group or condition effects in the LMMs were entered into a correlation analysis. Results were considered significant with *p* < 0.05 (uncorrected).

## 3. Results

### 3.1. Behavioral Data

#### 3.1.1. FA Rate and Classical RT Analysis

An overview of all behavioral performance estimates per group and condition is given in Table 2. A significant effect of condition on FA rates was identified. Higher FA rates were found in the visual compared to the auditory condition (*F*(1, 59) = 60.43, *p* < 0.001, *R*^2^ = 0.51). Additionally, a group effect was revealed (*F*(2, 58) = 3.64, *p* = 0.03, *R*^2^ = 0.11). Both children with HA (*p_adj_* = 0.06, *d* = −0.85) and children with ADHD (*p_adj_* = 0.07, *d* = −0.63) showed marginally higher FA rates than the TD group, while children with ADHD did not differ from children with HAs (*p_adj_* = 0.56, *d* = 0.22). Furthermore, male participants showed trend wise higher FA rates than female participants (*F*(1, 58) = 3.41, *p* = 0.07, *R*^2^ = 0.06).

Next, using classical RT_go_ analyses, a group-by-condition interaction was identified (*F*(2, 59) = 3.23, *p* = 0.047, *R*^2^ = 0.10). All groups showed lower RTs, i.e., faster reactions, in the visual than the auditory condition (HA: *F*(1, 14) = 76.38, *p* < 0.001, *p_adj_* < 0.001, *R*^2^ = 0.85; ADHD: *F*(1, 19) = 48.94, *p* < 0.001, *p_adj_* < 0.001, *R*^2^ = 0.72; TD: *F*(1, 26) = 165.18, *p* < 0.001, *p_adj_* < 0.001, *R*^2^ = 0.86). While groups did not differ in the visual condition (*F* < 1), a trend wise group effect on RTs was found in the auditory condition (*F*(2, 58) = 3.81, *p* = 0.03, *p_adj_* = 0.06, *η*^2^ = 0.12). That is, children with ADHD showed faster reactions than TDC (*p_adj_* = 0.02, *d* = 0.81), but did not differ from children with HAs (*p_adj_* = 0.29, *d* = 0.38), which in turn were not distinguished from TDC (*p_adj_* = 0.29, *d* = 0.43).

Response variability (SD_RTgo_) was affected by condition (*F*(1, 59) = 6.64, *p* = 0.01, *R*^2^ = 0.10) and group (*F*(2, 58) = 3.20, *p* = 0.048, *R*^2^ = 0.10). RTs were more variable in the auditory than the visual condition. The ADHD group showed a marginally higher response variability than the TD group (*p_adj_* = 0.06, *d* = −0.86), while the HA group did not differ from the ADHD (*p_adj_* = 0.60, *d* = −0.22) nor TD (*p_adj_* = 0.18, *d* = −0.63) group.

No significant SAT was identified for the two groups of interest (HA: auditory: *r_s_* = −0.06, *p* = 0.42, visual: *r_s_* = −0.04, *p* = 0.44; ADHD: auditory: *r_s_* = −0.16, *p* = 0.25, visual: *r_s_* = −0.28, *p* = 0.11). TDC did not show a significant SAT in the visual condition (*r_s_* = −0.27, *p* = 0.09), but only in the auditory condition (*r_s_* = −0.40, *p* = 0.02).

#### 3.1.2. Maximum Likelihood Estimation of RTs

The comparison of the parameter estimates of the of the ex-Gaussian distribution yielded a significant effect of condition (*F*(1, 59) = 118.90, *p* < 0.001, *R*^2^ = 0.67) and group (*F*(2, 58) = 6.43, *p* = 0.003, *R*^2^ = 0.18) on *mu*. *Mu* was smaller in the visual than in the auditory condition. Post-hoc comparisons of the group effect showed that children with ADHD reacted faster (lower *mu*) than the TDC (*p_adj_* = 0.003, *d* = 1.07). RTs of the HA group were marginally lower than the RTs of the TD group (*p_adj_* = 0.05, *d* = 0.75) and did not differ from the ADHD group (*p_adj_* = 0.39, *d* = 0.32). The group-by-condition interaction effect on *mu* was trend wise significant (*F*(2, 59) = 2.83, *p* = 0.07, *R*^2^ = 0.09). This was due to the fact that the group effect was largely driven by the auditory (*F*(2, 58) = 6.75, *p* = 0.002, *p_adj_* = 0.005, *η*^2^ = 0.19), in contrast to the visual condition (*F*(2, 58) = 2.13, *p* = 0.13, *p_adj_* = 0.13, *η*^2^ = 0.07). Furthermore, according to sigma values of the ex-Gaussian distribution, response variability was higher in the auditory than the visual condition (*F*(1, 59) = 39.12, *p* < 0.001, *R*^2^ = 0.40). Finally, for *tau* scores of the ex-Gaussian RT distribution, i.e., prolonged reactions, a main effect of condition (*F*(1, 59) = 11.46, *p* = 0.001, *R*^2^ = 0.16) and group (*F*(2, 58) = 4.59, *p* = 0.01, *R*^2^ = 0.14) was observed. All children showed higher *tau* scores in the auditory than the visual condition. Children with ADHD showed more inattentive responses than TDC (*p_adj_* = 0.02, *d* = −0.99). The HA group did not differ from the ADHD group (*p_adj_* = 0.75, *d* = −0.13) and were marginally more inattentive than the TD group (*p_adj_* = 0.05, *d* = −0.85).

### 3.2. FNIRS Analyses

#### 3.2.1. Functional Brain Activation

First, possible modality-specific (auditory, visual) changes of HbO and HbR were inspected per group (HA, ADHD, TD) by means of heat maps, depicting the neural activation of each condition contrasted against baseline (see Figure 3 and Figure 4 for the heat maps, and the Appendix A for the respective *t*-tests). Descriptively, the activation maps of both children with HAs and children with ADHD differed from TDC: The HA group showed broader activation in the temporal cortex during the auditory condition and higher activity in the right inferior and left superior frontal cortex during the visual condition than TDC. Children with ADHD showed more diffuse activation patterns than TDC, with higher parietal activity and broader frontal activity, yet, lower, less focal recruitment of the right inferior prefrontal gyrus (IPFG).

#### 3.2.2. Effects on Changes in HbO and HbR

For the ΔHbO, the following effects on neural activity were revealed: three Chs above the left dorsolateral prefrontal cortex (DLPFC) and the right frontopolar cortex showed higher ΔHbO in the visual than the auditory condition across groups. Contrary, higher ΔHbO in the auditory compared to the visual condition were revealed in the left pre-motor and supplementary motor area (pre/SMA), the left IPFG and right superior temporal cortex (STG). Moreover, the ADHD group showed higher ΔHbO than the TD group in the right supramarginal gyrus and the right primary somatosensory cortex. Only during the auditory condition, was a higher activation apparent in the ADHD group compared to the TD and HA group in the left supramarginal gyrus. Finally, while TDC showed no modality-specific activation in the right temporopolar region, the HA group showed higher ΔHbO in the auditory than the visual condition.

Comparable to the ΔHbO, 4 Chs above the left DLPFC and right temporal areas revealed lower ΔHbR, thus likely demonstrating higher activation, during the auditory than the visual condition. Additionally, the ADHD group showed higher ΔHbR, thus likely lower activation, in bilateral parietal, and left temporal as well as frontal regions than the HA group, and lower activation than the TD group in the left STG and bilateral frontal areas, independent of modality. Finally, only during the auditory condition, children with ADHD showed higher ΔHbR, lower activation, than the HA and TD group in the right DLPFC. All the effects on ΔHbO/ΔHbR are listed in Table 3 and Table 4 and illustrated in Appendix A, respectively.

#### 3.2.3. Brain-Behavior Correlations

Brain-behavior correlations were revealed for the TD and ADHD group (see Appendix A). For the TD group, a lower activation of the left DLPFC was associated with elevated FA rates in the auditory condition. In the visual condition, higher HbR in TDC, likely reflecting lower activation, in the right DLPFC was associated with an increased FA rate. For the ADHD group, an increase in HbR, lower activation, in the right medial temporal gyrus (MTG) and the left somatosensory cortex was associated with a higher FA rate in the visual condition. No significant correlations were found for the HA group.

## 4. Discussion

The objective of the current study was to elucidate the neural basis of auditory and visual response inhibition in children with HL, fitted with bilateral HAs, and children with ADHD. In particular, it was examined whether a shared dysfunctional supra-modal or modality-specific neural activation could be identified that contributes to the inattentive and impulsive behavior compared to TDC.

Generally, all children showed a higher FA rate and faster RTs in the visual compared to the auditory condition. In the present study, the response inhibition to nogo trials was thus superior in the auditory condition. Notably, reactions to auditory stimuli were however prolonged and highly variable. Previous studies similarly reported a high inconsistency and variability in responses to auditory stimuli, as reflected by an elevated omission rate and RT variability [46,47].

Despite the explicit exclusion of children with HAs with a diagnosis of ADHD, parental/teacher ratings of ADHD symptoms in the HA group were between the TD and ADHD group. In line with this, though contrary to our hypothesis, the current auditory and visual go/nogo task revealed that children with HAs did not differ from children with ADHD in their RTs or FA rates. Both the HA and ADHD groups showed faster, more impulsive reactions than TDC. Compared to the TDC, reactions were particularly faster during the auditory condition. The HA and ADHD groups, unlike the TDC, did not show a SAT in the auditory condition and therefore probably did not adjust their RTs as effectively in order to reduce FAs. While descriptively, FA rates for the HA group were particularly elevated in the auditory and for the ADHD group in the visual condition, both groups showed higher FA rates than the TDC across modalities. In addition, children with ADHD showed more attention lapses than TDC, as indicated by the ex-Gaussian parameter *tau*, independent of modality. Attention lapses of the children with HAs did not differ from the ADHD group either. The observed impulsive behavioral responses and elevated parental ADHD symptom ratings of the HA group are consistent with previous neuropsychological studies of children with HL [25,29]. Despite potentially different pathophysiological mechanisms, this suggests that sensory deprivation could lead to phenotypically similar behavioral deviations from TDC as ADHD. These alterations, in response inhibition, appear to be present across sensory modalities. Yet, differences between auditory and visual response inhibition were observed, even in TDC, and point to modality-specific characteristics of response inhibition processes. The modality-specific efficiency might potentially underlie the asynchronous development of vision and audition [47,72]. This might possibly explain the decrease in differences between auditory and visual RTs with increasing age, which has been previously shown in TD individuals [73]. The behavioral differences, in turn, might underlie the neural development as well as a potentially different neural organization of auditory and visual response inhibition mechanisms.

Although the current neural findings should be inspected with caution due to the exploratory nature of the (uncorrected) analyses, the current study, to our knowledge, was the first to provide tentative indications of modality- and group-specific neural activations. While across groups, higher activity was detected in the temporal regions likely indicating auditory processing, differences between modalities beyond the temporal cortices were found, such as within the frontal cortex and the pre/SMA. The frontostriatal network, including the (inferior) frontal cortex and pre/SMA, plays a crucial role in response inhibition [1]. The modality-specific activation in these regions might underlie differences in connectivity of the sensory systems to the response inhibition network. Comparable modality-specific differences in response inhibition have been previously reported between the tactile and visual modality in adults [8,74], suggesting that behavioral and neural processes involved in response inhibition depend on the modality of the presented stimulus. In contrast to the tactile system that has more direct connections to the frontostriatal network, the auditory and visual systems might differently connect to frontal regions via an indirect path over the parietal association hubs [75,76]. While many (attention) mechanisms in TD might generally operate in a supra-modal manner (e.g., see Deng, et al. [77] for a current study on shared auditory and visual spatial attention processes), the current results hint to the sensory-specific efficiency of response inhibition. Notably, while exploratory analyses suggested that a decreased activation of left and right DLPFC cortex was associated with an increased FA rate in TDC in the auditory and visual condition respectively, the ADHD group showed an association between lower activity in temporal and somatosensory areas and an increased FA rate during the visual condition.

Next to these modality-specific changes in HbO and HbR, the first indications of activation patterns specific to children with ADHD and specific to children with HL with bilateral HAs were found. While the ADHD group showed higher and less focal activation of right supramarginal and somatosensory cortices, lower activation in the left STG and bilateral frontal regions was observed. The observed effects were independent of stimulus modality. In particular, the largest effect, observed in the left IPFG (pars opercularis), and effects in other frontal regions that indicated higher ΔHbR, thus likely lower activation, in children with ADHD than TDC are in agreement with previous studies reporting a lower (right inferior) frontal activity in ADHD during response inhibition paradigms [35,78,79,80,81]. The IPFG, in particular, as part of the frontostriatal response inhibition network, has been linked to response inhibition and an action stopping-mechanism [82].

Contrary to the ADHD group, the activation patterns of the HA group differed only from the TDC in the right temporopolar cortex. That is, children with HAs showed a modality-specific activation with higher activation during the auditory than the visual condition, whereas TDC showed a modality-independent activation pattern. The elevated activation during the auditory condition might reflect a higher effort required due to the HL to achieve a comparable performance level. Albeit poor response inhibition was observed in the HA group independent of modality, no supra-modal alteration in the activation patterns of the children with bilateral HL fitted with HAs were found. Yet, based on the current fNIRS probe configuration, we cannot rule out changes in neural activation entirely, since the amount of Chs utilized for the assessment was limited (Section 4.1). Further, while descriptively a higher activation of right inferior frontal and left superior frontal cortices during the visual condition compared to TDC can be seen, which might suggest higher effort or result of altered neural connectivity (e.g., [19]), these (uncorrected) baseline contrasts should be interpreted with caution. While children with HAs and children with ADHD thus showed indifferent behavioral responses, a differential functional disorganization specific to each group was revealed.

### 4.1. Limitations and Future Applications

The current study provides first insights into the differences between auditory and visual response inhibition on behavioral and neural level across different pediatric samples. Hereby, fNIRS proved to be a suitable technique to investigate the current (auditory) task under a naturalistic and quiet experimental setting.

The presented results, however, should be considered in the context of some limitations. Due to strict time constraints, we assessed auditory and visual response inhibition within one paradigm to allow both a direct comparison of the two modalities, and the comparison of the two groups of interest to TDC. This did not allow an event-related neural analyses and a contrast of cortical activation during nogo (successful/not successful) vs. go (successful/not successful) trials, due to the insufficient number of total trials. However, if response inhibition is considered the dominant mechanism underlying mixed go/nogo blocks [40,83], the underlying neural basis of the nogo process should be revealed when a go/nogo block is contrasted against a low-level baseline. Nevertheless, future event-related approaches of a similar paradigm could offer further insights into visual and auditory response inhibition. Event-related designs and simultaneous fNIRS-EEG assessments might add important information on modality-dependent temporal aspects of response inhibition. Second, we cannot rule out that differences between modalities resulted from the different stimulus characteristics of the current auditory and visual stimuli. Irrespective of differing stimulus characteristics, auditory processing seems to occur in a similar time frame to visual processing [84], and previous studies utilizing simple RT tasks have shown that RTs to auditory stimuli are faster than RTs to visual stimuli [85,86]. Thus, if response inhibition efficiency could be supra-modal, auditory RTs in a response inhibition task should be faster than, or at least as fast as, visual RTs. This suggests that the modality-specific effects reported herein were likely specific to response inhibition and are driven by modality-specific effects of inhibitory control rather than differences in the speed of processing an auditory vs. a visual stimulus. An alternative explanation could be that the observed modality-specific effects resulted from differences in the distinction between the go and nogo stimuli. It might have been easier to distinguish between the auditory nogo stimulus and the auditory go stimuli than it was to distinguish between the visual nogo stimulus and the visual go stimuli. Although previous studies provided comparable evidence of a modality-specific response inhibition efficiency in TD adults and children [8,73], future studies might investigate response inhibition with different visual and auditory stimuli to validate the current findings. Analogously, the order of auditory and visual conditions has not been counterbalanced, which might have potentially primed children to the first presented modality [87]. Finally, it is important to note that the degree of hearing (loss), the age at HL and HA amplification, as well as HA use, might have affected performance. For example, a previous study indicated that cortical re-organization depends on the degree of HL [88], which in turn is likely to affect behavioral performance. The current sample of children with HAs showed a wide variety with respect to their degree of HL and no pure-tone audiometry was performed for TDC and children with ADHD. Normal hearing was based on the mandatory early hearing screening (U9, including headphone-based audiometry) and parental report. Future studies should aim to assess (i) an extended paradigm version with a higher number of trials, (ii) different and counterbalanced presentation of auditory and visual stimuli, and (iii) the effect of the degree of hearing and HL etiology, as well as HA use and amplification within larger samples.

## 5. Conclusions

The current study provided first evidence for differences in response inhibition in children with HL and children with ADHD compared to TDC across and between the auditory and visual modality. Notably, children with HAs showed similar behavioral performance alterations in the go/nogo task to children with ADHD, suggesting that early sensory deprivation might put children at risk of developing response inhibition deficits. On the neural level, however, group-specific altered activation patterns compared to TDC were revealed, which might underlie the observed behavioral changes. While children with HL fitted with HAs mainly showed differences in neural activity during the auditory condition, children with ADHD showed alterations across modalities. The current study emphasizes the importance of neuropsychological investigations of visual and auditory response inhibition in children with HL. Furthermore, while the current study suggests that the observed behavioral difficulties appear not to be related to ADHD, the behavioral similarities between HL and ADHD warrant a careful screening of ADHD symptoms in children with HL to detect behavioral difficulties (in)dependent of ADHD, which consequentially, should be considered in rehabilitation programs. In addition, the current findings carry important implications for future ADHD investigations. Further studies examining complementary auditory to already existing visual paradigms in ADHD are warranted, as behavioral and neural alterations in response inhibition were found in both modalities within the current study. Replications with larger samples and different combined visual and auditory response inhibition paradigms are encouraged.

## Figures and Tables

**Figure 1 brainsci-10-00307-f001:**
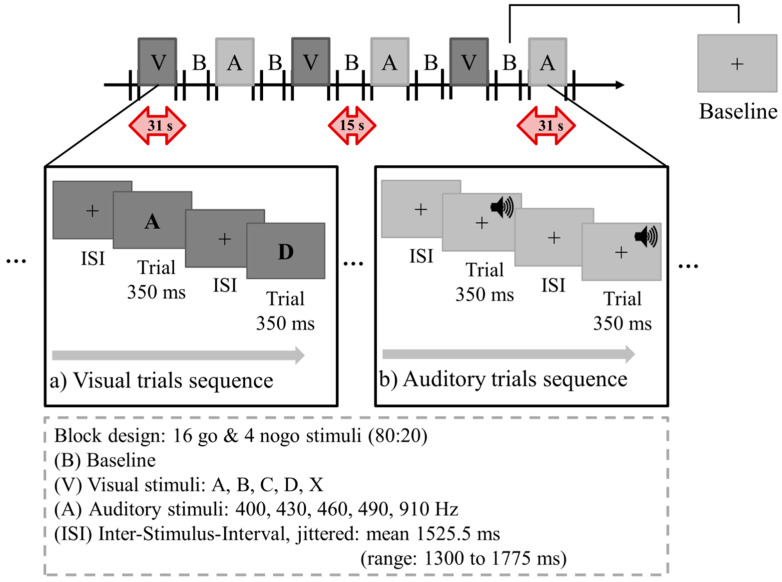
Experimental design and trial structure. Abbreviations: B, Baseline; V, Visual; A, Auditory; ISI, Inter-stimulus-interval.

**Figure 2 brainsci-10-00307-f002:**
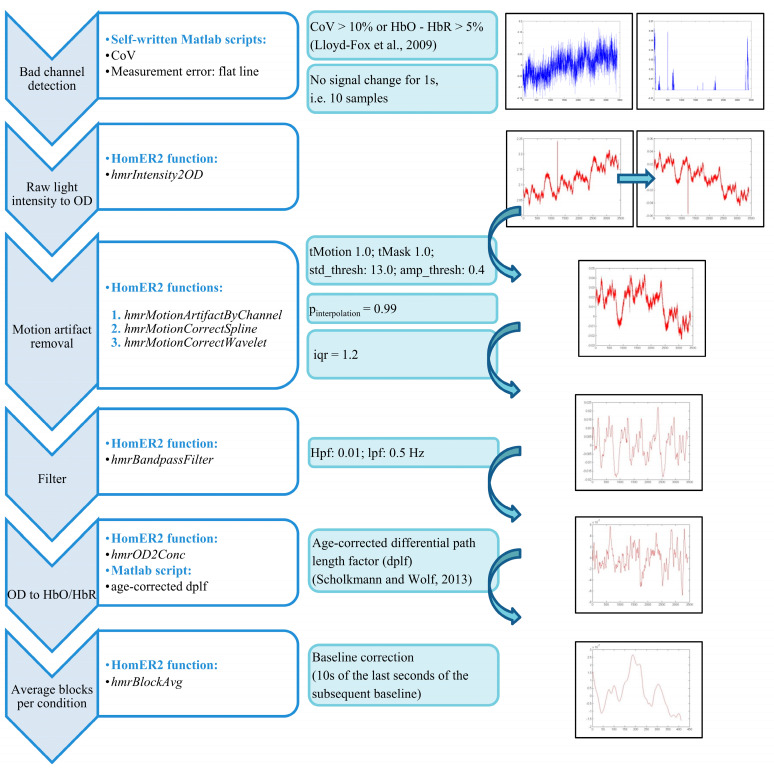
FNIRS preprocessing pipeline. Abbreviations: CoV, coefficient of variation; HbO, oxygenated hemoglobin; HbR, deoxygenated hemoglobin; OD, optical density; lpf, low-pass filter; hpf, high-pass filter; dplf, differential path length factor.

**Figure 3 brainsci-10-00307-f003:**
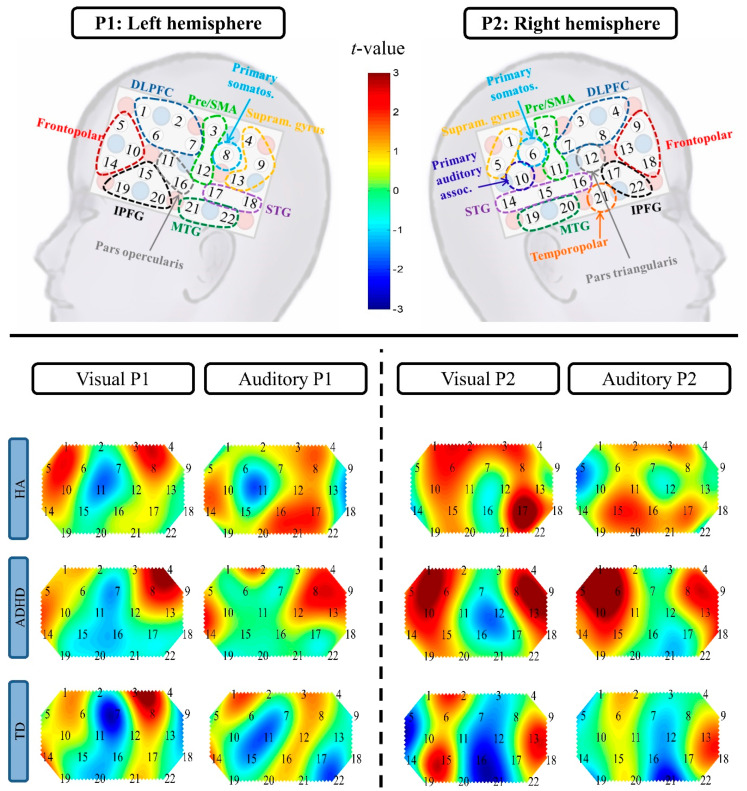
Heat maps (*t*-maps) of the ΔHbO contrast against baseline per group (HA, ADHD, TD) and condition (auditory, visual). The respective t-values can be found in the Appendix A. Abbreviations: P, Probe set; DLPFC, dorsolateral prefrontal cortex; pre/SMA, pre-motor and supplementary motor area; STG, superior temporal gyrus; MTG, medial temporal gyrus; IPFG, inferior prefrontal gyrus; primary somatos., primary somatosensory cortex; primary auditory assoc., primary auditory association cortex; supram. gyrus, supramarginal gyrus; HA, hearing aid; ADHD, attention deficit/hyperactivity disorder; TD, typically developing.

**Figure 4 brainsci-10-00307-f004:**
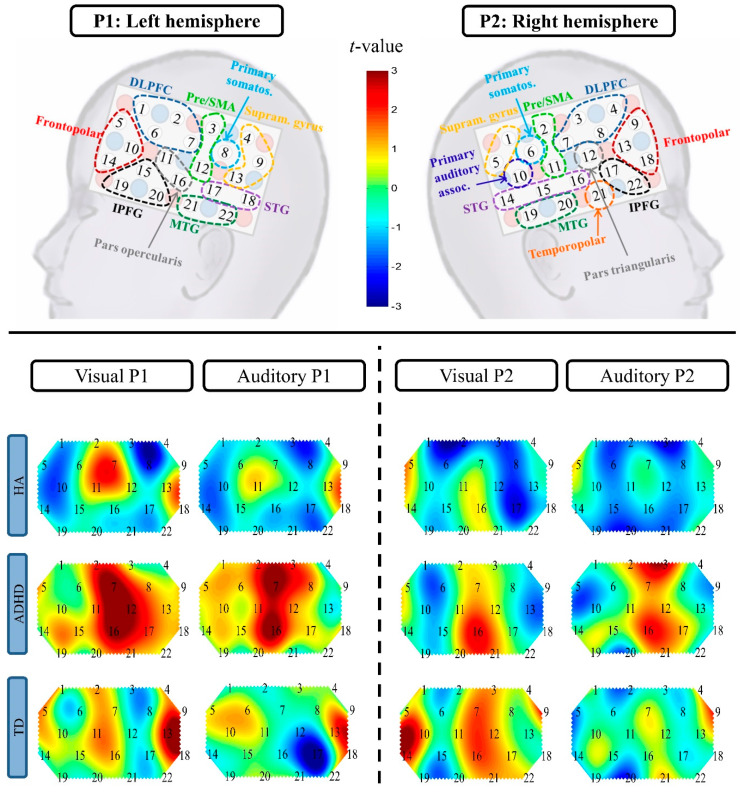
Heat maps (*t*-maps) of the ΔHbR contrast against baseline per group (HA, ADHD, TD) and condition (auditory, visual). The respective t-values can be found in Appendix A. Abbreviations: P, Probe set; DLPFC, dorsolateral prefrontal cortex; pre/SMA, pre-motor and supplementary motor area; STG, superior temporal gyrus; MTG, medial temporal gyrus; IPFG, inferior prefrontal gyrus; primary somatos., primary somatosensory cortex; primary auditory assoc., primary auditory association cortex; supram. gyrus, supramarginal gyrus; HA, hearing aid; ADHD, attention deficit/hyperactivity disorder; TD, typically developing.

**Table 1 brainsci-10-00307-t001:** Demographic data for the behavioral and neural analyses.

		HA		ADHD		TD		Group Comparison		Post hoc	
		*M*	*(SD)*	*M*	*(SD)*	*M*	*(SD)*	*χ^2^/F*	*p*		
Behavioral analyses analysis Behavioral behavioral task	*n*	*n* = 15		*n* = 20		*n* = 27					
	Age	9.23	(1.93)	10.25	(2.01)	9.87	(1.77)	*F*(2,59) = 1.25		0.29	
	CBCL Inattention	59.50	(6.43)	64.00	(7.81)	52.85	(4.70)	*F*(2,57) = 18.27		<0.001	TD < HA < ADHD
	FBB-HKS Total	61.83	(26.84)	77.24	(23.49)	42.05	(29.91)	*F*(2,47) = 7.99		0.001	TD < ADHD (and TD < HA (*p_adj_* = 0.07))
	FBB-HKS Inattention	72.29	(22.82)	85.20	(19.93)	48.29	(30.06)	*F*(2,52) = 11.51		<0.001	TD < HA and TD < ADHD
	FBB-HKS Hyperactivity	69.75	(18.23)	76.16	(21.51)	49.62	(20.10)	*F*(2,49) = 9.22		<0.001	TD < HA and TD < ADHD
	FBB-HKS Impulsivity	57.80	(22.07)	75.40	(23.18)	45.67	(26.43)	*F*(2,53) = 7.79		0.001	TD < ADHD (and HA < ADHD (*p_adj_* = 0.06))
	Gender (m:f)	5:10		14:6		17:10		*χ*^2^(2, *n* = 62) = 5.20		0.07	
Neural analyses	*n*	*n* = 12		*n* = 15		*n* = 15					
	Age	9.11	(1.73)	10.12	(2.07)	10.06	(1.84)	*F*(2,39) = 1.31		0.33	
	Inattention CBCL (*t*)	60.25	(6.62)	66.53	(7.20)	52.07	(3.54)	*F*(2,38) = 21.05		<0.001	TD < HA < ADHD
	FBB-HKS Total	63.30	(25.67)	82.92	(19.47)	40.73	(30.69)	*F*(2,31) = 8.27		0.001	TD < ADHD (and TD < HA (*p* = 0.07) < ADHD (*p* = 0.07))
	FBB-HKS Inattention	72.64	(22.75)	89.87	(12.87)	49.27	(32.08)	*F*(2,34) = 10.00		0.001	TD < ADHD and TD < HA (and HA < ADHD (*p* = 0.07))
	FBB-HKS Hyperactivity	70.40	(17.33)	81.00	(20.77)	47.00	(18.11)	*F*(2,33) = 10.21		<0.001	TD < HA and TD < ADHD
	FBB-HKS Impulsivity	54.83	(23.36)	82.67	(21.19)	49.27	(26.61)	*F*(2,35) = 7.79		0.002	TD < ADHD and HA < ADHD
	Gender (m:f)	5:7		9:6		10:5		*χ*^2^(2, *n* = 62) = 5.20		0.07	

Note: The demographic data is shown separately for the behavioral analyses, that included children who completed the behavioral task, and the neural analyses, that included children who completed the behavioral task and had valid neural data. Demographic data and t-values/percentiles for the inattention scales of the CBCL (Child Behavior Checklist) and the FBB-HKS scale (Fremdbeurteilungsbogen für Hyperkinetische Störungen) are depicted for the HA (hearing aid), ADHD (attention deficit/hyperactivity disorder) and TD (typically developing) group.

**Table 2 brainsci-10-00307-t002:** Summary of the behavioral performance scores per group and condition.

Variable	Modality	HA (*n* = 15)		ADHD (*n* = 20)		TD (*n* = 27)		Group Comparison			
		*M*	*SD*	*M*	*SD*	*M*	*SD*	*F*	*p_adjusted_*	Effect Size *(R^2^/η^2^)*	Post hoc *(p_adjusted_, d)*
FA rate	Visual	37.22	15.06	39.17	17.54	29.63	16.56	*F*(2, 58) = 3.64	0.03	0.11	ADHD > TD (0.07, −0.63)
(%)	Auditory	20.56	24.17	17.92	18.79	9.57	13.22				HA > TD (0.06, −0.85)
RT_go_	Visual	432.48	89.86	436.64	77.99	436.46	69.15	*F* < 1	-	-	-
	Auditory	611.62	125.49	576.05	96.51	636.91	69.55	*F*(2, 58) = 3.81	0.03 (0.06)	0.12	ADHD < TD (0.02, 0.81)
SD_RTgo_	Visual	121.88	63.23	141.70	79.31	110.58	41.89	*F*(2, 58) = 3.20	0.048	0.10	ADHD > TD (0.06, −0.86)
	Auditory	187.75	64.24	184.25	70.92	143.77	42.88				
Mu	Visual	319.37	70.58	304.18	59.24	348.42	85.05	*F*(2, 58) = 6.43	0.003	0.18	ADHD < TD (0.003, 1.07)
	Auditory	457.86	122.61	424.08	119.11	538.66	91.90				HA < TD (0.05, 0.75)
Sigma	Visual	43.52	28.36	38.65	26.38	52.15	26.28	*F* < 1	-	-	-
	Auditory	92.99	56.28	79.61	47.59	82.46	39.75				
Tau	Visual	106.77	59.29	126.46	76.67	82.50	43.28	*F*(2, 58) = 4.59	0.01	0.14	ADHD > TD (0.02, −0.99) HA > TD (0.05, −0.85)
	Auditory	152.96	92.27	152.38	80.21	100.53	52.85				

Note: Values for RTgo and SDRTgo are shown in raw data format (ms) for illustrative purposes, while for the analyses, the log-transformed values were utilized. Abbreviations: HA, hearing aid; ADHD; attention deficit/hyperactivity disorder; TD, typically developing; FA, false alarm; RT, reaction time.

**Table 3 brainsci-10-00307-t003:** Effects of group and condition on ΔHbO.

Channel	Brain Region	Statistics	Direction of Effect(s)
**Condition effect**			
P1 Ch6	left DLPFC	*F*(1, 31) = 5.28, *p* = 0.03, *R*^2^ = 0.15	visual > auditory
P1 Ch7	left DLPFC	*F*(1, 37) = 7.48, *p* = 0.01, *R*^2^ = 0.17	visual > auditory
P2 Ch13	right frontopolar cortex	*F*(1, 36) = 5.19, *p* = 0.03, *R*^2^ = 0.13	visual > auditory
P1 Ch12	left pre/SMA	*F*(1, 39) = 4.65, *p* = 0.04, *R*^2^ = 0.11	auditory > visual
P1 Ch16	left pars opercularis	*F*(1, 36) = 4.93, *p* = 0.03, *R*^2^ = 0.12	auditory > visual
P1 Ch20	left IPFG	*F*(1, 32) = 6.40, *p* = 0.02, *R*^2^ = 0.17	auditory > visual
P2 Ch16	right STG	*F*(1, 36) = 7.88, *p* = 0.008, *R*^2^ = 0.18	auditory > visual
**Group effect**			
P2 Ch1	right supramarginal gyrus	*F*(2, 34) = 3.65, *p* = 0.04, *R*^2^ = 0.18	ADHD > TD (*p_adj_* = 0.03, *d* = −1.59)
P2 Ch 5	right supramarginal gyrus	*F*(2, 33) = 5.01, *p* = 0.01, *R*^2^ = 0.23	ADHD > TD (*p_adj_* = 0.01, *d* = −1.49), ADHD > HA (*p_adj_* = 0.06, *d* = −1.27)
P2 Ch 6	right primary somatosensory cortex	*F*(2, 38) = 3.94, *p* = 0.03, *R*^2^ = 0.17	ADHD > TD (*p_adj_* = 0.03, *d* = −1.22)
**Condition × Group effect**			
P1 Ch9	left supramarginal gyrus	*F*(2, 35) = 3.32, *p* = 0.048, *R*^2^ = 0.16	group effect in the auditory condition (*F*(2, 35) = 3.60, *p* = 0.04, *p_adj_* = 0.07, *η*^2^ = 0.17) ADHD > TD (*p_adj_* = 0.048, *d* = −0.86) ADHD > HA (*p_adj_* = 0.048, *d* = −1.01)
P2 Ch21	right temporopolar cortex	*F*(2, 33) = 6.31, *p* = 0.005, *R*^2^ = 0.28	condition effect for the HA group: auditory > visual (*p =* 0.004, *p_adj_* = 0.01, *R*^2^ = 0.66); condition effect for the ADHD group:auditory < visual (*p =* 0.04, *p_adj_* = 0.06, *R*^2^ = 0.33)

Abbreviations: P, probe set; Ch, channel; DLPFC, dorsolateral prefrontal cortex; pre/SMA, pre-motor and supplementary motor area; IPFG, inferior prefrontal gyrus; STG, superior temporal gyrus; TD, typical developing; HA, hearing aid; ADHD, attention deficit/hyperactivity disorder.

**Table 4 brainsci-10-00307-t004:** Effects of group and condition on ΔHbR.

Channel	Brain Region	Statistics	Direction of Effect(s)
**Condition effect**			
P1 Ch7	left DLPFC	*F*(1, 37) = 7.90, *p* = 0.008, *R*^2^ = 0.18	auditory < visual
P2 Ch14	right STG	*F*(1, 37) = 4.34, *p* = 0.04, *R*^2^ = 0.11	auditory < visual
P2 Ch16	right STG	*F*(1, 36) = 4.88, *p* = 0.03, *R*^2^ = 0.12	auditory < visual
P2 Ch20	right MTG	*F*(1, 38) = 10.57, *p* = 0.002, *R*^2^ = 0.22	auditory < visual
**Group effect**			
P1 Ch8	left primary somatosensory cortex	*F*(2, 37) = 4.88, *p* = 0.01, *R*^2^ = 0.21	ADHD > HA (*p_adj_* = 0.01, *d =* −1.40)
P1 Ch16	left pars opercularis	*F*(2, 36) = 6.79, *p* = 0.003, *R*^2^ = 0.27	ADHD > HA (*p_adj_* = 0.004, *d =* −1.64), ADHD > TD (*p_adj_* = 0.01, *d =* −1.21)
P1 Ch17	left STG	*F*(2, 39) = 4.62, *p* = 0.02, *R*^2^ = 0.19	ADHD > HA (*p_adj_* = 0.02, *d =* −1.04), ADHD > TD (*p_adj_* = 0.02, *d =* −0.98)
P2 Ch2	right pre/SMA	*F*(2, 36) = 3.87, *p* = 0.03, *R*^2^ = 0.18	ADHD > HA (*p_adj_* = 0.03, *d =* −1.28)
P2 Ch17	right IPFG	*F*(2, 32) = 3.41, *p* = 0.045, *R*^2^ = 0.18	ADHD > TD (*p_adj_* = 0.05, *d =* −0.18), HA < TD (*p_adj_* = 0.05, *d =* 1.10)
**Condition × Group effect**			
P2 Ch3	right DLPFC	*F*(2, 31) = 4.13, *p* = 0.03, *R*^2^ = 0.21	group effect in the auditory condition (*F*(2, 31) = 6.55, *p* = 0.004, *p_adj_* = 0.008, *η*^2^ = 0.19): ADHD > HA (*p_adj_* = 0.008, *d =* −1.47), ADHD > TD (*p_adj_* = 0.01, *d =* −1.11)

Abbreviations: P, probe set; Ch, channel; DLPFC, dorsolateral prefrontal cortex; STG, superior temporal gyrus; MTG, medial temporal gyrus; pre/SMA, pre-motor and supplementary motor area; TD, typical developing; HA, hearing aid; ADHD, attention deficit/hyperactivity disorder.

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
