# Peer review of "Auditory and Visual Response Inhibition in Children with Bilateral Hearing Aids and Children with ADHD"

_brainsci, 2020, doi:10.3390/brainsci10050307_

Round 1

Reviewer 1 Report

Manuscript ID: brainsci-772897

Summary
The manuscript Auditory and Visual Response Inhibition in Children with Bilateral Hearing Aids and Children with ADHD is an empirical study that investigates visual and auditory response inhibition using a Go Nogo paradigm on children with ADHD, hearing aids (HA) and typically developing children (TDC). Group comparisons were made on both on behavioral (task performance) and neural (brain) activation using functional near-infrared spectroscopy (fNIRS). Reported findings were: reaction times were faster and false alarm rate was higher in visual go-nogo as compared to auditory for all groups; Children with ADHD and HA reacted faster and made more commission errors as compared to TDC. Overall behavioral were similar for the two patient groups as compared to the TDC group. fNirs data showed different brain activation patterns in the patient groups despite similar behavioral performance, where the ADHD group displayed more distributed supra modal activation as compared to HA children. From this was concluded that sensory deprivation might cause inhibition deficits. However, the different brain activation patterns in ADHD vs. HA may instead indicate that the inhibition impairments may be due to these.

Overall comment
This is a well conducted empirical study that addresses important issues in the field of sensory perception in general an in particular in children with sensory deprivation and attention deficits/ADHD. A thorough and relevant background to the RQ’s are provided and methods, analytical strategy, and results are well described with figures and tables. Here I have to highlight that my knowledge in fNIRS methods and protocols are rather limited so this review will not go into detail with technical matters regarding this. Results and assessments are thoroughly presented and drawn conclusions are plausible. Despite that I have a number of concerns and objections regarding results and conclusions that will be presented below, in particular regarding the fNIRS part. The concerns starts with a three major points regarding design and participant information and thereafter a section with six minor points.

Major points

1) when you are talking about sensory perception and executive functioning it is very surprising that nothing is said about hearing among the ADHD group, apart from mentioning that they had normal hearing. Is that self-, teacher, or parent reported? Is that valid for both ears? Moreover, we neither receive any information about hearing in the TDC group. I hope that it’s possible to include data on hearing from all participants in participant characteristics. (the HA groups hearing data are very well documented in graphs).

2) The experimental design, if, as stated, differences between visual and auditory performance is expected it is kind of surprising why not condition order is counter balanced. Now all test sessions start with visual stimuli and is followed after 15 seconds by auditory stimuli. The first condition will serve as a primer to the latter, that is the brain uses a forward model to produce a template and predictions can be made in advance. Therefore, the learning history of the participant is (or could be) relevant, in this case effects condition order cannot be excluded, see e.g. (Sanmiguel, Widmann, Bendixen, Trujillo-Barreto, & Schroger, 2013). I am aware of that this cannot be changed in retrospect, but it can be mentioned in the discussion as a possible limitation.

3) The temporal solution of fNIRS is comparable with the one of fMRI, therefore I am a bit puzzled why not Go-trials are separated from successful Nogo trials. Now we receive the overall hemodynamic response during the experiment, and from this it is not feasible to draw conclusions about hemodynamic responses of inhibition. See for example (Steele et al., 2013). As I see it, a key interest of the present study was to investigate if there are any hemodynamic differences in inhibition responses between children with ADHD and hearing loss that are not modality specific. Is there a more general or global response inhibition response that is similar for these two groups as suggested by (Frank, 2006). These data are available but as stressed in the limitations the number of nogo trials are not sufficient. This calls for a total reframing of the fNIRS part of the present study, the hemodynamic response is limited go-performance, and this should be mentioned in the introduction. Moreover, all nogo trials should be removed from analyses while they are confounding results. Finally, due this I suggest that the results from fNIRS to be toned down a bit.

Minor points
4) The supra modal organization of attentional networks is interesting; I think it’s worth to present the default mode network in the introduction. ADHD is proposed to suffer from an inability to suppress DMN (Liddle et al., 2011). There is also in the DMN context suggested that the brain has a superior extrinsic task mode that is more or less similar irrespective of task, see e.g. (Hugdahl, Raichle, Mitra, & Specht, 2015).

5) In line 57, where audiovisual integration is presented the orienting response/reflex is worth mentioning. I don’t know whether this is altered in patients with hearing loss – add a sentence about that.

6) in the abstract and in line 85 it is mentioned that ADHD patients show impaired concentration in noisy environments without any references. This really depends, under certain circumstances in certain tasks noise can be beneficial for persons with ADHD e.g. (Söderlund, Björk, & Gustafsson, 2016; Söderlund, Sikström, & Smart, 2007).

7) Effect sizes, due to the small differences between groups indicated by high p-values these could be added to Table 2 with effect sizes in Cohen’s d for all significant results. In particular when results with p-values over .05 are presented as trends.

8) FDR corrections for multiple comparisons were made - will significance remain using the more conservative Bonferroni correction?

9) The fact that the HA group did display longer reaction times in auditory processing is by no means surprising. This could reflect impaired top down processing (as required in inhibition) or the alternate explanation, they didn’t perceive the target signal properly due to their hearing loss. I think this should be mentioned in the discussion.

References.
Frank, M. J. (2006). Hold your horses: a dynamic computational role for the subthalamic nucleus in decision making. Neural Netw, 19(8), 1120-1136. Retrieved from http://www.ncbi.nlm.nih.gov/entrez/query.fcgi?cmd=Retrieve&db=PubMed&dopt=Citation&list_uids=16945502

Hugdahl, K., Raichle, M. E., Mitra, A., & Specht, K. (2015). On the existence of a generalized non-specific task-dependent network. Front Hum Neurosci, 9, 430. doi:10.3389/fnhum.2015.00430

Liddle, E. B., Hollis, C., Batty, M. J., Groom, M. J., Totman, J. J., Liotti, M., . . . Liddle, P. F. (2011). Task-related default mode network modulation and inhibitory control in ADHD: effects of motivation and methylphenidate. J Child Psychol Psychiatry, 52(7), 761-771. doi:10.1111/j.1469-7610.2010.02333.x

Sanmiguel, I., Widmann, A., Bendixen, A., Trujillo-Barreto, N., & Schroger, E. (2013). Hearing Silences: Human Auditory Processing Relies on Preactivation of Sound-Specific Brain Activity Patterns. 33(20), 8633-8639. doi:10.1523/jneurosci.5821-12.2013

Söderlund, G. B. W., Björk, C., & Gustafsson, P. (2016). Comparing Auditory Noise Treatment with Stimulant Medication on Cognitive Task Performance in Children with Attention Deficit Hyperactivity Disorder: Results from a Pilot Study. Frontiers in Psychology, 7(1331). doi:10.3389/fpsyg.2016.01331

Söderlund, G. B. W., Sikström, S., & Smart, A. (2007). Listen to the noise: noise is beneficial for cognitive performance in ADHD. Journal of child psychology and psychiatry, and allied disciplines, 48(8), 840-847. Retrieved from http://www.ncbi.nlm.nih.gov/pubmed/17683456

Steele, V. R., Aharoni, E., Munro, G. E., Calhoun, V. D., Nyalakanti, P., Stevens, M. C., . . . Kiehl, K. A. (2013). A large scale (N=102) functional neuroimaging study of response inhibition in a Go/NoGo task. Behavioural Brain Research, 256, 529-536. doi:10.1016/j.bbr.2013.06.001

Reviewer 2 Report

This study investigated performance on a Go/No Go task, which tapped into attention to both visual and auditory stimuli, and corresponding neural activity in a group of children with hearing loss, Attention Deficit Hyperactivity Disorder (ADHD) and typically developing (TDC) controls. Overall, I think this manuscript is well written with comprehensive detail. Work that attempts to elucidate common and distinct cognitive and neural processes among different clinical groups is highly valuable in helping us understand these conditions. A particular strength was the inclusion of three groups, having TDC in addition to the two target groups. There are several areas where the manuscript could be improved, which are outlined below.

Introduction:

  1. Page 2, lines 59-61: Here and in multiple places throughout the manuscript the authors suggest that children with HL show behavioural difficulties and atypical neural activity. However, it would be helpful to be specific and to give a more detailed profile of the types of behaviour patterns observed among children with HL. Given that they are being compared to an ADHD group, it would also be helpful to describe if there is an overlap in the types of behaviour problems. It may also be interesting to indicate the prevalence of ADHD among children with HL, if the patterns of behaviour are quite similar, it may mean that children with HL are also more likely to be diagnosed with ADHD, strengthening the need to investigate the underpinnings of the two conditions to see if they truly do have overlap beyond the behavioural level.
  2. Page 2, lines 66-67: Here, and throughout the introduction, the authors discuss different variables from the Go/No Go task. Here the False Alarm (FA) variable is discussed and later on the authors talk about reaction time variability, omission rates, etc. It may be helpful to have a brief section here to describe the key variables drawn from the Go/No Go task and what they are so that the reader can refer back to this when the key findings are described later on.
  3. Page 2, line 92: The manuscript investigates differences in auditory and visual modalities in the Go/No Go task and draws on previous research, which has found that children with ADHD show more pronounced difficulty when the task taps into visual modality. Is there any potential explanation or hypothesis for why this may be the case?
  4. Page 3, lines 109-111: Similar to the point above, can the authors hypothesise any differences between the HL and ADHD groups on the Go/No Go task when the task is auditory and when it is visual?
  5. Page 3, lines 111-112: The hypothesis about the organisation of attention networks of children with HL and ADHD compared to TDC is very vague. Do the authors expect the HL and ADHD groups to differ from each other as well and in what ways are they expected to differ?

Materials and Methods:

  1. I struggled to follow the procedure outlined here. In particular because there is no explicit statement saying that the behavioural (Go/No Go) task and fNIRS were done concurrently and also because Table 1 breaks down the demographic data into the behavioural and neural tasks. This made it seem like there were two tasks, a purely behavioural one and an additional task done with fNIRS. I think it would be helpful to include a statement on page 5, line 169 to simply say that fNIRS was being recorded during the task. Additionally, I would rename the two sections of Table 1 (Behavioural and Neural) to indicate that the first section is data for all participants who completed the behavioural task and that the second section is data for the participants who completed the behavioural task and had valid neural data.
  2. Page 5, line 156: Can the authors justify why they used a single task to measure both the auditory and visual Go/No Go, as opposed to separate tasks for each modality? It seems that this reduced the number of trials for each modality. It may also make the task more challenging because the participants have to remember which type of trial they are getting, in addition to the target response.
  3. Page 7, line 239: This is a minor detail, the authors say that they excluded trials where RTs were below 100ms. Did they exclude any trials with RTs that were too long?

Results:

  1. Overall, I think that the results are clearly written, but there is quite a lot to follow. Perhaps it would be helpful to include a summary of the group differences in Table 2 in addition to the Ms and SDs.
  2. Given that the authors are doing quite a few comparisons between three groups that are relatively small in size, it may be helpful to include effect size in addition to the p-values.

Discussion:

  1. Page 13, lines 362-370: The authors report that their findings suggest that all groups performed better on the auditory condition. This is quite an interesting finding and it would be nice to give some indication as to why this may be the case. It is particularly interesting given that the HA group showed this same pattern, so being born with hearing loss did not change this pattern for them.
  2. Page 14, line 383: The authors draw a parallel between the behavioural patterns observed in the HA group and their performance on the Go/No Go task, but this is not something that is actually tested. I was surprised that the authors did not look at associations between ADHD symptoms and task performance/neural activity in the HA group. I understand that this is an additional analysis in a small sample, but it could perhaps be added as Supplementary Material. The issue is that presently the authors are drawing on similarities between behavioural and cognitive performance without knowing if they are actually related, particularly as the fNIRS data suggests that the neural patterns in the HA group are different from those in ADHD.

Reviewer 3 Report

The present manuscript describes an interesting fNIRS study on response inhibition tested by means of a go/nogo-task in children with hearing loss (HL) but supplied with hearing aids, in children suffering from attention-deficit-hyperactivity-disorder (ADHD), as well as typically developing children (TDC). Authors compared response inhibition in auditory and visual stimuli.

The introduction is concise and clearly written. Authors provide solid motivations for conducting this study from both a thematic and methodological point of view.

The methods and analyses are sound. However, I have some suggestions to be addressed in these sections:

  1. Please add subjects’ handedness. I assume that subjects were right-handed and responded to go trials with the right hand. This is important as a mixing of handedness might alter lateralization effects.
  2. Please add in Table S1 also the etiology of hearing loss as well as the age of HL diagnosis, not only the age of first hearing add fitting. If etiology and/or age of HL diagnosis should differ, please discuss a potential influence on the found results in the discussion section.
  3. With respect to fNIRS data analysis, authors report uncorrected values due to the large amount of fNIRS channels. I understand the problem but I would like to suggest performing statistical analyses on regions of interests (ROIs), selected based on expectations from previous literature. In such a case, the reduced amount of ROIs compared to channels could allow to integrate in the analyses also factors such as region (e.g., anterior to posterior) and hemisphere (left vs. right) which could provide additional statistical clarity about potential topographical differences across groups. And with this limited amount of ROIs, authors may be able to correct their data for multiple comparisons.

Results are reported in a clear and comprehensible fashion and are supported by clear Tables and Figures. Please add the unity of color-coding in Figures 3 and 4. Is this mmol/l?

The discussion is well-written and comprehensible. What I missed was a discussion on a potential developmental influence on the found (especially neural) effects. Could you please provide some information from previous literature on what was found or expected in HL, ADHD, and TD adults? Were similar activation patterns found in auditory and/or visual go/nogo tasks or other inhibitory tasks (such as Stroop, Flanker task etc.) in adults compared to children? Can we assume that children between 6 and 13 years of age behave neurally like adults? Please provide some discussion in regard.

Reviewer 4 Report

I greatly enjoyed reading this interesting paper. The topic is important and the experimental procedures were carried out expertly. I was especially impressed with the rigorous manner in which the data processing and analyses were conducted. That being said, I do have a few comments and suggestions to further improve this paper.

Introduction:

  • On page 2, the authors write “a supra-modal organization of attention networks and therefore likely response 52 However, the utilization, i.e. brain regions and the amount of involvement, of these 53 networks appears to vary dependent on the modality and features attended [9].” In simpler words, this seems to say that attention network activation depends on attention, which is a circular argument.
  • Line 65: how does early auditory deprivation alter the education of the child?
  • Lines 98-112: The hypotheses for fNIRS findings are not specified in the introduction, other than the statement “a differential functional disorganization of attention networks” was expected. The behavioral hypotheses are slightly better described—behavioral difficulties in response inhibition in children with HL and children with ADHD (compared with TD children?) but worse for ADHD children. In its current form, the paper seems to be a replication of tasks that have already been done before in these populations, but just not in the same study using this combination of methods. I think a stronger focus on the specific hypotheses up front would help to clarify the novel contribution of the study.

Methods:

  • Line 136: I am curious about why you asked children with ADHD to refrain from their medications before participation. What are the temporary effects of skipping their medications? Does this have any side consequences—e.g., mood—that could alter performance? It is curious to me that ADHD children were unmedicated, but the HL children were wearing their devices. This decision seems asymmetrical to me. Some further explanation/motivation would help.
  • Please consider adding a bit more information about the HL sample, such as age at amplification and years of HL use. These factors have been shown to play a crucial role in language and cognitive development. Are either of these significantly correlated with the behavioral or neural DVs?
  • Line 211: Please explain the “stringent matching procedure” a bit more. As the groups are not actually matched on gender, a few more details about this procedure would be helpful.

Results:

  • One of the advantages of fNIRS is that is provides good temporal and spatial resolution. I understand that the authors did not have specific hypotheses regarding the time course of the neural responses, but it seems a shame to not at least perform some exploratory analyses on the temporal aspect of the response.
  • The authors report both Hbo and HHb results. My understanding is that it is more common to only report one (e.g., Hbo; Lloyd-Fox, 2010) as the two signals are interdependent anyway. Is it necessary in this case to report both? I feel that focusing on one signal (and perhaps moving the other to the supplementary) would help to streamline and simplify the results section, which would make the paper easier to read and understand.

Discussion

  • Lines 362-370: I find this paragraph to be slightly confusing. The authors state initially that children performed better in the auditory condition, but later on that they were more inattentive and slower (which sounds, to me, like worse performance). Perhaps this could be slightly reworded to be less ambiguous.
  • Lines 386-389: The authors note the condition (auditory vs. visual) differences even in TD children and interpret this as revealing response inhibition differences based on modality. As auditory and visual inputs are processed at different speeds, can you rule out differences in the speed of perceptual processing of the input between auditory vs. visual modalities? Could this play a role in the children’s reaction times, in addition to modality-specific response inhibition? I see that the authors address this point in lines 445-455, but I am not able to fully follow their argument. Some clarification would be helpful.
  • The authors suggest in their conclusion that the neural findings complement their behavioral findings, but I struggle to see a strong correspondence between the two. Some more explanation in the discussion of how the neural findings match the behavioral findings is needed to make this connection clearer.
  • Line 458: in suggesting future directions, the authors bring up “degree of HL”. This seems to come up out of the blue, without any prior mention of the degree of HL and task performance. Some additional explanation in the discussion would help to motivate this suggestion.
  • Line 469: The authors also propose that their findings suggest that HL children should also be screened for ADHD symptoms in clinical practice. The reasoning for this is not obvious to me, as they explicitly excluded any children with both HL and ADHD from the current study (as they should have) and the findings do not appear to suggest that the behavioral findings in the HL group are due to underlying ADHD.

Minor:

  • Page 2 line 46: “Typically developing (TD; [7])” is incomplete
  • Line 445: “Second, albeit previous study comparably hint towards a modality-specific efficiency results in TD adults and children” – this sentence doesn’t make grammatical sense.
  • Line 429: “discerned” should be changed to “interpreted”
  • Final comment: the English writing is very good, but as a native speaker, for me at times the language usage is a bit strange. One suggestion is to have a native speaker proofread the paper to read more smoothly.

Round 2

Reviewer 1 Report

Comment to Authors

The authors have addressed all the concerns I raised thoroughly and I thank them for their honesty and the down toning of the neural findings. I still have my doubts about these findings, but as the results are presented and discussed right now this is transparent and up to the reader to conclude and judge the dignity of the presented findings. As stated by authors, that this is a pilot study I am looking forward to a replication the study with an improved design were all the shortcomings are taken care of.

Kind regards
Reviewer 1
